# Performance of Modified Alumina-Supported Ruthenium Catalysts in the Reforming of Methane with $CO_2$

Silvia Carolina Palmira Maina, Irene María Julieta Vilella, Adriana Daniela Ballarini and Sergio Rubén de Miguel *

Instituto de Investigaciones en Catálisis y Petroquímica "Ingeniero José M. Parera" (INCAPE), Facultad de Ingeniería Química, Universidad Nacional del Litoral—CONICET, Centro Científico Tecnológico CONICET Santa Fe (CCT-SF), Predio CONICET "Dr. Alberto Cassano" Colectora Ruta Nacional N° 168 km. 0, Pje. El Pozo, Santa Fe 3000, Argentina
* Correspondence: sdmiguel@fiq.unl.edu.ar

**Abstract:** Ruthenium (1 wt%) catalysts supported on alumina doped with alkaline (Na and K) and alkaline earth metals (Ba, Ca, and Mg) of different concentrations (1, 5, and 10 wt%) were tested in the dry reforming of methane. All catalysts were prepared by the successive impregnation method. Supports were characterized by X-ray diffraction, BET surface area, temperature-programmed desorption of $CO_2$, and 2-propanol dehydration. Additionally, catalysts were characterized by temperature-programmed reduction (TPR), temperature-programmed oxidation (TPO), transmission electron microscopy (TEM), and X-ray photoelectron spectroscopy (XPS). Stability tests to study coke deposition were performed using long-time dry reforming reactions. All the catalysts showed good catalytic activity, and activity falls were never detected. Ru metallic phase seemed to be resistant to coke formation even though its particles are sintered during a long-term reaction.

**Keywords:** Ru catalysts; doped alumina; syngas; dry reforming





## 1. Introduction

Syngas is produced almost exclusively by steam-reforming of natural gas (SR) [1].

This reaction was widely studied over Ni catalysts that give high reaction rates at low cost when compared with noble metal catalysts [2–4]. SR has two disadvantages. On the one side, the highly endothermic reaction proceeds at high water/methane volume ratios that demand a lot of energy. In this case, a high $H_2/CO$ molar ratio useful for subsequent processes, such as alcohol or ammonia production, is obtained [5,6]. On the other side, carbon deposits on the catalyst can be reduced with an excess of steam (2.5 to 4 moles of steam per mole of methane) [7].

Dry reforming (DR) has been proposed as one of the most promising technologies for harnessing these two greenhouse gases [2,8–10] and has turned out to be a good alternative. In this case, the reforming of natural gas is carried out with carbon dioxide:

$$CH_4 + CO_2 \rightarrow 2\,H_2 + 2\,CO \qquad \Delta H^\circ = 247.3\ kJ/mol \qquad (1)$$

This reaction gives a low $H_2/CO$ ratio [11] compared to steam reforming, which is useful for Fischer Tropsch, methanol [12], oxo alcohols [13], acetic acid, and dimethyl ether syntheses [14]. A relevant advantage is the low cost of both methane and carbon dioxide that would be transformed into high-value-added products [7], thus decreasing their emissions and impact on the greenhouse effect [15]. Additionally, the reaction represents a very convenient energy transfer system [16].

In this DR reaction, there is no water steam to clean up the carbon deposits from the catalyst surface [10,17]. The following reaction produces carbon deposits:

$$CH_4 \rightleftharpoons C + 2H_2 \qquad \Delta H^\circ_{298\,K} = 74.9\ KJ\ mol^{-1} \qquad (2)$$

$$2\,CO \rightleftharpoons C + CO_2 \qquad \Delta H^\circ_{298\,K} = -172.4\ KJ\ mol^{-1} \qquad (3)$$

Nevertheless, an industrial approach to this process has not been established due to the low activity and coke formation on the catalysts during the prolonged reaction times. A proper catalytic formulation should reduce carbon deposition over the catalyst (1) or in the cooling zones, at the exit of the reaction, according to reaction (2). It also has to be considered that because of its endothermic character [18], the reaction demands a lot of energy (combustion oven).

The support also plays an important role in the interaction between the metallic particles and the support being studied [19]. The addition of calcium, magnesium, or barium to alumina used as support for Ni catalysts has been successful, particularly magnesium-doped alumina ones that were both active and stable and reduced coke formation, such as in how Alipour et al. [20] reported. Furthermore, Juan-Juan et al. [21] found good catalytic activity and low coke deposition when Ni catalysts supported on potassium-doped alumina were used. Nevertheless, as there is a lack of reports about the use of Ba, Mg, and Ca as alumina promoters in noble metal-based catalysts for the dry reforming reaction, it is interesting to study the addition of alkaline earth metals to alumina.

The dry reforming reaction has been tested with both non-noble metals (Ni, Co, and Fe) [1,4,19,22–27] and noble ones (Pt, Ru, Rh, Re, Pd, Ir) supported with $Al_2O_3$, $SiO_2$, $ZrO_2$, MgO, and others [8,9,28–42]. Even though good performances have been found, this meaning good activities and selectivities, the problem that needs to be tackled is the catalyst deactivation caused by carbon deposits.

Noble metal-based catalysts showed high activity and low-carbon deposition for syngas production from dry reforming of methane [37]; however, Ni ones are cheaper and limitless as compared with the former ones. A very important challenge is to develop catalysts for industries that are highly resistant to carbon deposition in addition to high selectivity and activity.

This work aims to analyze both the catalytic performance and stability of Ru catalysts based on alumina doped both with alkaline and alkaline earth metals in the methane dry reforming reaction at 750 °C. The stability study of the Ru catalyst with better performance by the reaction–regeneration experiments at high temperatures and during very long reaction times (59 h) was carried out. In addition, a complete characterization of the metallic phase of different catalysts was performed by using different techniques.

## 2. Results and Discussion

### 2.1. Catalyst Characterization

#### 2.1.1. A/AE Modified Supports

Previous studies show a complete characterization of the support doped with different A/AE concentrations [38].

The supports doped with Ca, Ba, Na, and K and calcined at 800 °C kept the gamma alumina structure, showing traces of oxides, aluminates, and carbonates. Mg-doped alumina showed a phase modification as magnesium aluminate. Textural properties were slightly modified by 1 wt% A/AE addition, but the higher the A concentration, the lower the specific surface, while alkaline earth samples showed a slight decrease in this property.

The acid function of the supports was partially inhibited by Ba, Ca, and Mg, this effect increasing with the concentration. On the other hand, A metals completely poisoned the acid sites even at the lowest loadings. A/AE metal addition to alumina creates many basic sites of different natures whose concentration increases with the metal concentration. This effect was confirmed by the TPD test that showed wide peaks and/or shifts with maxima at higher temperatures (T = 200 °C) as well as the presence of new peaks at high temperatures (T = 400–600 °C) [38].

#### 2.1.2. Metallic Phase

Figures 1 and 2 show the TPR profiles of Ru/$Al_2O_3$-A/AE(1 wt%) and Ru/$Al_2O_3$-A/AE(1, 5, and 10 wt%) catalysts, respectively. From Figure 1, it can be seen that Ru/$Al_2O_3$

catalyst showed a reduction peak at about 220 °C. For the catalysts supported on A/AE doped alumina, the peak shifts to slightly lower temperatures, up to 200 °C for Ru/Al$_2$O$_3$-K catalyst. In addition, at 140–150 °C, a smaller reduction peak was also found for all the series, except for the undoped Ru/Al$_2$O$_3$ catalyst, due to easily-reduced ruthenium oxides that could be formed by the interaction between the metal and the doped supports (see Figure 1). Other authors [43] also found two reduction peaks for Ru/SiO$_2$ catalysts, a main one at 200 °C due to the reduction of RuO$_2$ particles and a secondary one at 150 °C produced by well-dispersed RuO$_x$ species. Moreover, Faroldi et al. [44] also found for Ru/Al$_2$O$_3$ two reduction peaks but at lower temperatures (140 and 100 °C) that were attributed to crystalline phases of supported RuO$_2$. The TPR profile of the alumina support (without Ru) did not show H$_2$ consumption at temperatures lower than 500 °C, which is in the zone of Ru reduction.

From quantification results of the areas corresponding to the different reduction profiles, it was determined that after the reduction treatment with hydrogen at high temperatures, most of Ru would be in a zerovalent state and only a small fraction would be oxidized, such as other authors found [45].

As Figure 2 shows, Ru catalysts supported on alumina doped with higher A/AE concentrations gave the main reduction peak at approximately 200 °C, with some modifications. The catalysts supported on Ba or Na-modified alumina showed similar peaks to the ones with 1 wt% A/AE metals, while the reduction peaks of samples supported on Ca or K-doped alumina were wider. In addition, Ru/Al$_2$O$_3$-Ca and Ru/Al$_2$O$_3$-Mg (5 and 10 wt%) catalysts showed wider and half-divided peaks due to the presence of larger Ru particles (observed by TEM, as it will be discussed later) and oxidized species on the support that would be reduced at higher temperatures, causing the shift of the peaks (see Figure 2).

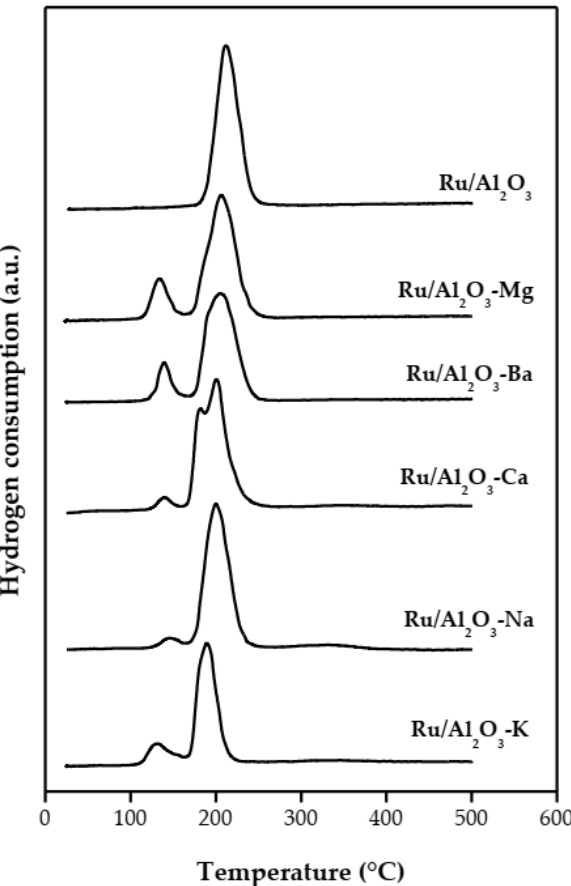

**Figure 1.** TPR profiles of Ru/Al$_2$O$_3$-A/AE(1 wt%) catalysts.

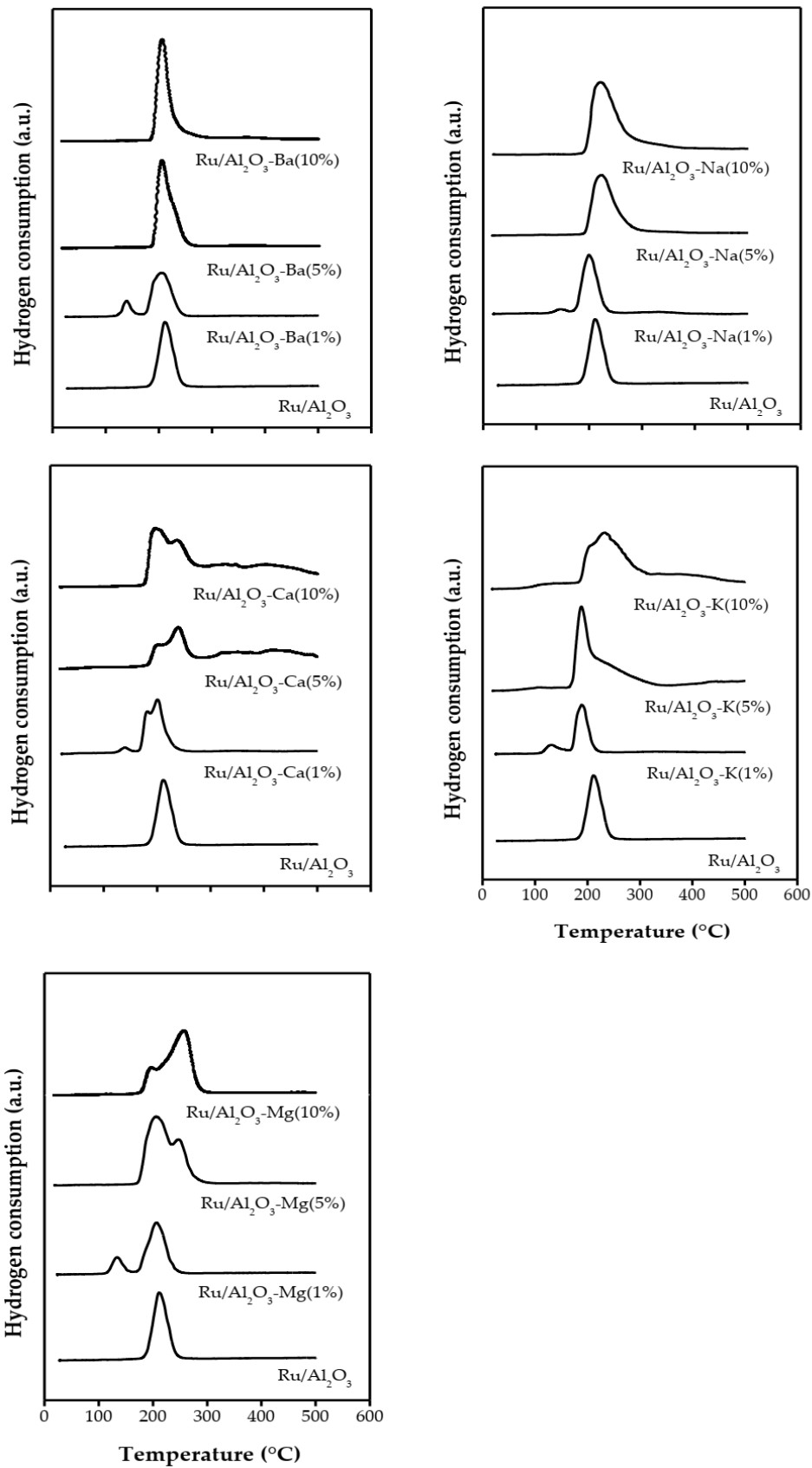

**Figure 2.** TPR profiles of Ru/Al$_2$O$_3$-A/AE(1, 5, and 10 wt%) catalysts.

No reduction signal at low temperatures was found in catalysts supported on alumina doped with high A/AE metal contents, as it happened for those supported on alumina doped with 1 wt% A/AE metals.

With respect to XPS experiments, the Ru3d signal was first analyzed using an asymmetric line with a 90/10 L/G ratio and a medium–wide/height ratio of 0.7. The spin-orbital overlapping was $4.1 \pm 0.05$ eV [45,46]. Because of carbon contamination, C1s signals overlapped Ru3d ones, these being of very low intensity. For this reason, $Ru3p_{3/2}$ signals were used to characterize the oxidation state of the catalysts. Figure 3a–c shows $Ru3p_{3/2}$ XPS spectra of $Ru/Al_2O_3$, $Ru/Al_2O_3$-Ca(1 wt%), and $Ru/Al_2O_3$-K(1 wt%) catalysts, which were deconvoluted into two signals, one at about 461 eV that was assigned to metallic Ru and the other one at about 464 eV, corresponding to $RuO_x$. [47,48]. Results show that Ru was in a zerovalent state (75–85%) and also forming oxidized species (15–25%) in all catalyst series. These results are consistent with TPR ones and also with those found by different authors. In this sense, Mazzieri et al. [49] found that for $Ru/Al_2O_3$ catalysts reduced at 400 °C, the $Ru^0$ content reached 85% for some catalysts. On the other hand, Elmasides et al. [45] found that on alumina, about 80% of Ru was reduced to a metallic state by hydrogen treatment at 550 °C, while on $TiO_2$, and the reduction was complete. Otherwise, Faroldi et al. [44] analyzed the behavior of $Ru/La_2O_3$ catalysts in the dry reforming reaction of $CH_4$ at 550 °C and found that after the reaction, Ru was mainly reduced to a metallic state, but there was a low proportion of oxidized Ru, both in trivalent and tetravalent states.

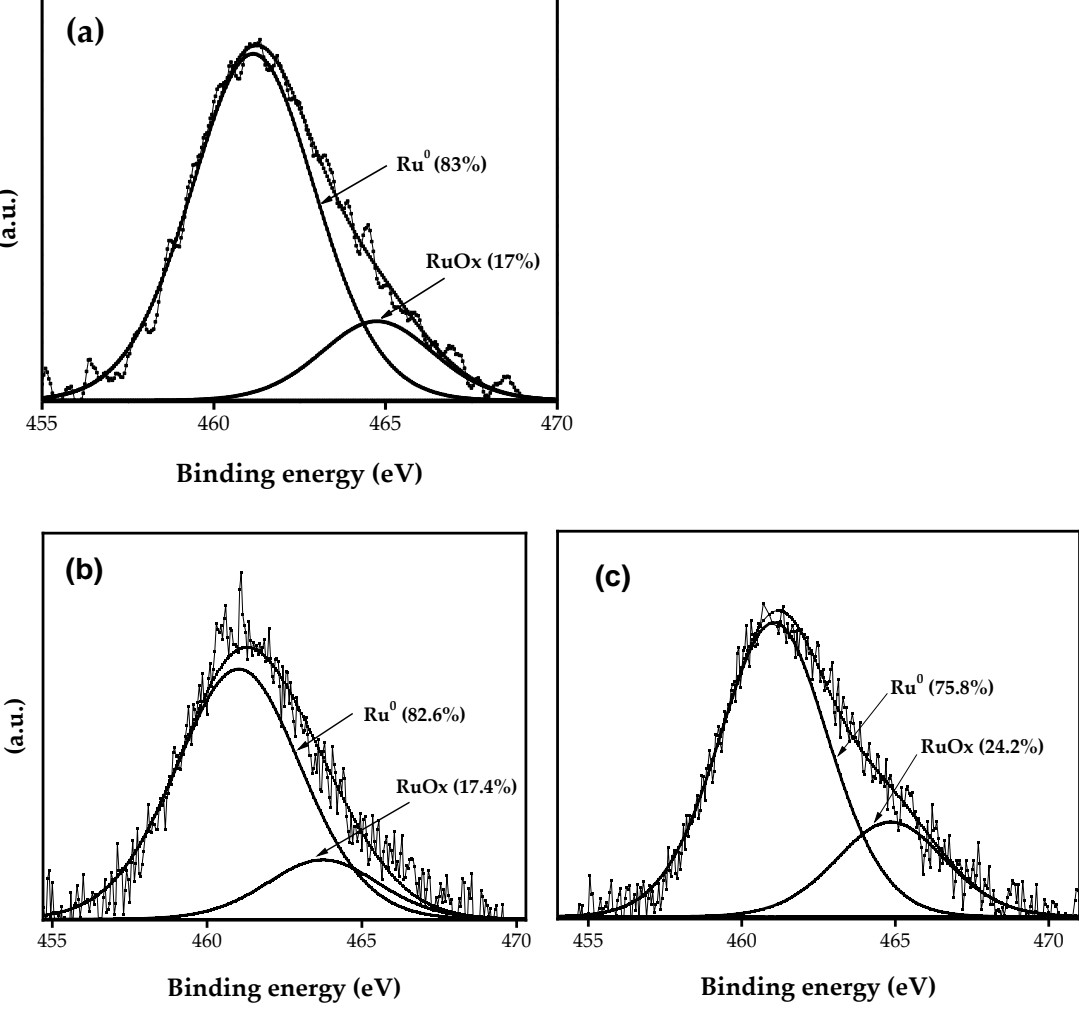

**Figure 3.** $Ru3p_{3/2}$ XPS signals for $Ru/Al_2O_3$ (**a**), $Ru/Al_2O_3$-Ca(1 wt%) (**b**), and $Ru/Al_2O_3$-K(1 wt%); (**c**) catalysts, indicating the percentage of reduced and oxidized species.

Figure 4a,b show XPS spectra ($3p_{3/2}$ signal) of Ru/Al$_2$O$_3$-Ca(10 wt%) and Ru/Al$_2$O$_3$-K(10 wt%) catalysts. The deconvolution of both spectra could be performed using the greatest peak corresponding to Ru$^0$ at 461.4–461.6 eV, which means that the Ru reducibility to metallic state and the second minor peak at 465.5 eV corresponds to RuOx at 465.5 eV. It means that the Ru reducibility to metallic state is about 93–100%. This indicates that the higher the A/AE metal concentration, the higher the Ru reducibility. It is the metallic Ru that forms new active sites that promote methane and CO$_2$ conversion in the dry reforming reaction for these catalysts, as will be seen later.

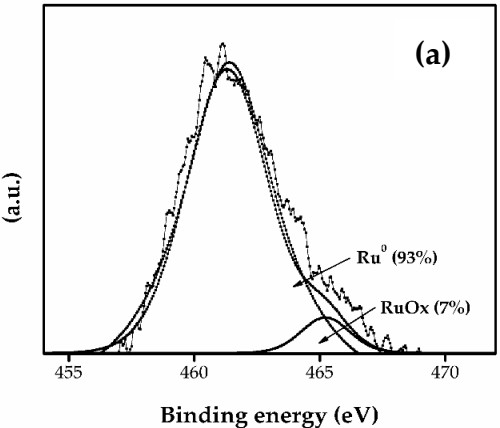
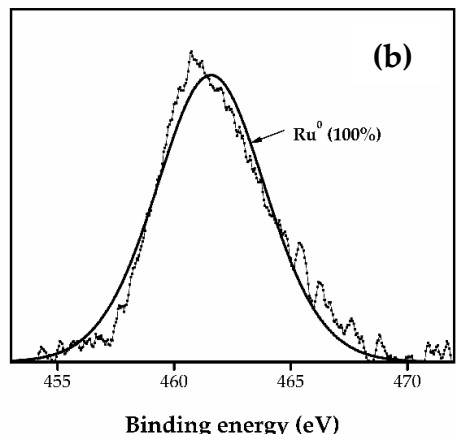

**Figure 4.** Ru$3p_{3/2}$ XPS signals for Ru/Al$_2$O$_3$-Ca(10 wt%) (**a**) and Ru/Al$_2$O$_3$-K(10 wt%) (**b**) catalysts.

Figures 5 and 6 show the particle size distributions for the different Ru catalysts supported on alumina doped with 1 and 10 wt% A/AE, respectively. In addition, Table 1 shows the average particle sizes of Ru/Al$_2$O$_3$-A/AE(1 wt%) and Ru/Al$_2$O$_3$-A/AE(10 wt%) catalysts. The TEM microphotographs of these catalysts are shown in the Appendix A (Figures A1–A9).

The average particle sizes of Ru particles of the monometallic catalyst and also of the series doped with 1 wt% A/AE metal were similar (between 1.5 nm and 1.75 nm), these showing very narrow particle size distributions, as seen in Figure 5. Table 1 shows that the average particle sizes of Ru were, in all cases, higher for the samples doped with 10 wt% than for the ones with 1 wt% A/AE metals. Additionally, Figure 6 indicates that Ru particle size distributions in Ru/Al$_2$O$_3$-A/AE(10 wt%) catalysts were much wider. Moreover, it can be observed that the catalysts supported on Al$_2$O$_3$ doped with alkali metals (10 wt%) had smaller average particle sizes than those doped with AE metals (10 wt%). In this sense, it must be pointed out that, among the samples with a high load of A/AE, the Ru/Al$_2$O$_3$-K (10 wt%) catalyst showed the smallest mean particle size (2.8 nm).

**Table 1.** Average metallic particle size (d) of Ru catalysts deposited on alumina doped with 1 and 10 wt% A/AE metals.

| Catalyst | d (nm) | |
|---|---|---|
| | x = 1 wt% | x = 10 wt% |
| Ru/Al$_2$O$_3$-Ba(x) | - | 5.2 |
| Ru/Al$_2$O$_3$-Ca(x) | - | 4.7 |
| Ru/Al$_2$O$_3$-Mg(x) | 1.6 | 5.7 |
| Ru/Al$_2$O$_3$-Na(x) | 1.5 | 4.3 |
| Ru/Al$_2$O$_3$-K(x) | 1.6 | 2.8 |

Ru/Al$_2$O$_3$: d = 1.7 nm.

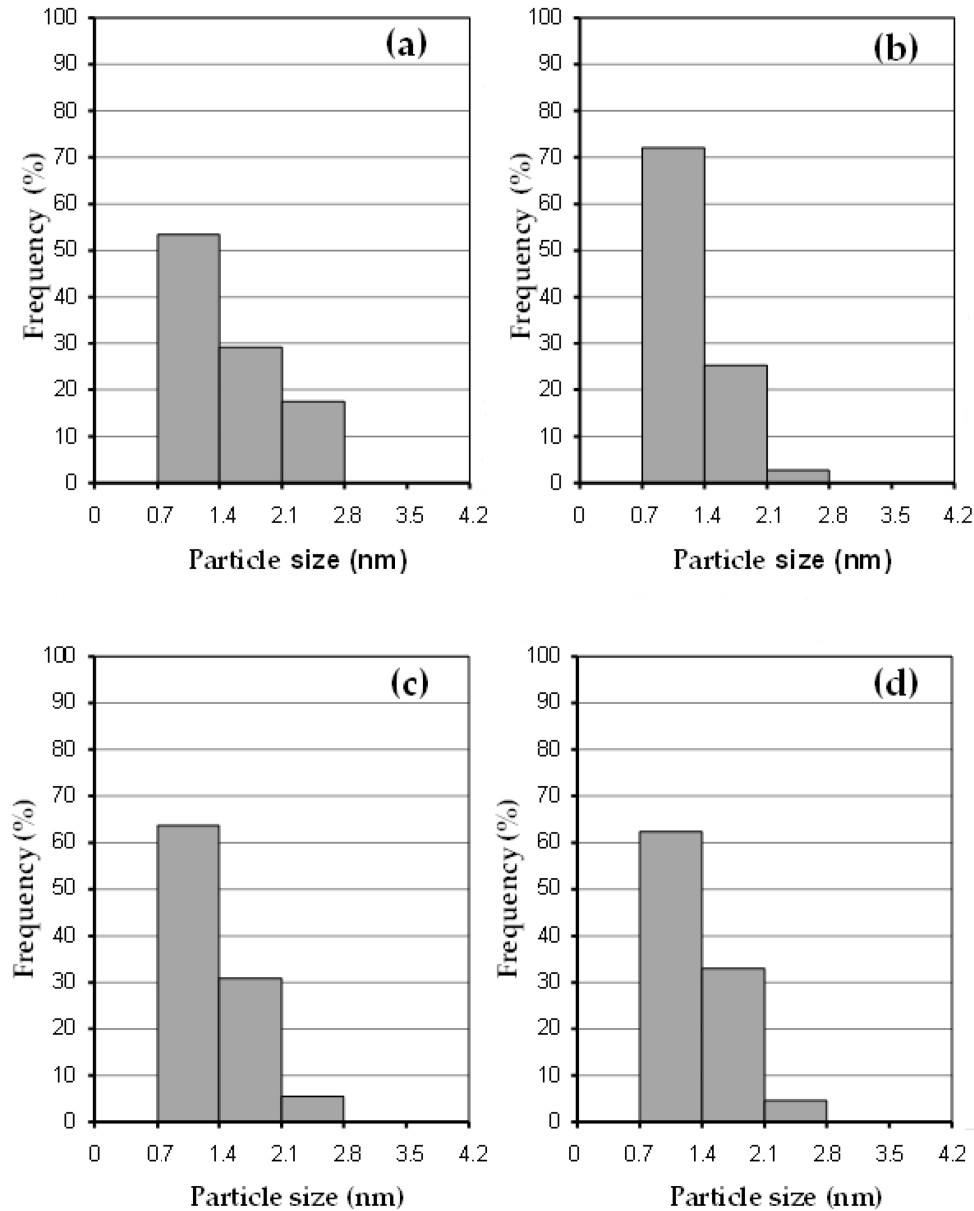

**Figure 5.** Particle size distributions for the different Ru/Al$_2$O$_3$-A/AE(1 wt%) catalysts. Without A/AE (**a**), A: Na (**b**), A: K (**c**), and AE: Mg (**d**).

*2.2. Dry Reforming Test of Ru Catalysts*

2.2.1. Influence of the Reduction Time

For all the catalysts, it was found that the shorter the reduction time (from 5 to 2 h), the lower the conversion of both reactants (CH$_4$ and CO$_2$). It would also seem that the longer the reduction time, the higher the metal reducibility. In order to study whether this effect was due to the heat treatment or to the presence of H$_2$, an experience using the same catalyst weight but thermally treated with He for 3 h after a reduction with H$_2$ for 2 h, was performed. Low conversions, similar to the ones with the hydrogen reduction for 2 h, were found. This indicates that the conversion decrease is caused by the reduction treatment with hydrogen and not by a thermal effect. Since Ru needs about 5 h reduction time in hydrogen to attain the highest activity, a 5 h-reduction time was used for all reaction experiments.

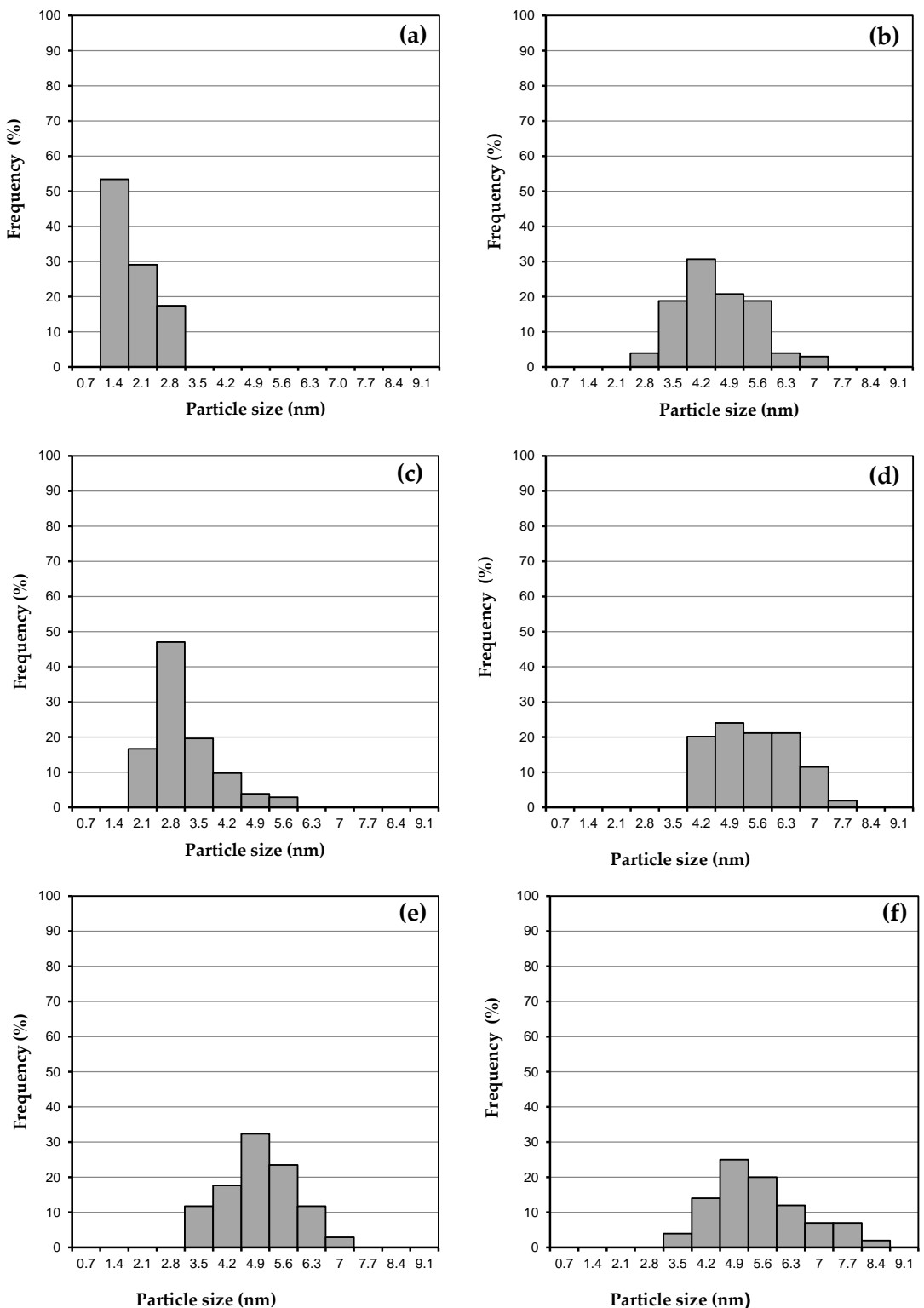

**Figure 6.** Particle size distributions for the different Ru/Al$_2$O$_3$-A/AE(10 wt%) catalysts. Without A/AE (**a**), A: Na (**b**), K (**c**), AE: Ba (**d**), Ca (**e**), and Mg (**f**).

2.2.2. Study of the Catalysts in Conditions Close to the Equilibrium State

Ru/Al$_2$O$_3$-A/AE(1 wt%) catalysts reduced at 750 °C for 5 h in situ were tested in a continuous methane dry reforming equipment to compare activity and selectivity values in conditions close to equilibrium by using a large catalyst weight (200 mg). Table 2 shows CH$_4$ and CO$_2$ conversions and H$_2$/CO molar ratios for Ru catalysts supported on 1 wt%

A/AE metal-doped alumina. In these conditions (close to equilibrium state), the different catalysts supported on doped and undoped alumina showed similar behavior, both in activity and selectivity ($H_2$/CO molar ratio).

**Table 2.** % $CH_4$ conversion, % $CO_2$ conversion, and $H_2$/CO molar ratio for Ru/$Al_2O_3$-A/AE (1 wt%) catalysts (750 °C, 135 min reaction time). Catalyst weight: 200 mg.

| Catalyst | X $CH_4$ (%) | X $CO_2$ (%) | Molar Ratio $H_2$/CO |
|---|---|---|---|
| Ru/$Al_2O_3$ | 62.4 | 86.1 | 0.45 |
| Ru/$Al_2O_3$-Ba(1 wt%) | 67.3 | 89.3 | 0.48 |
| Ru/$Al_2O_3$-Ca(1 wt%) | 62.7 | 87.4 | 0.48 |
| Ru/$Al_2O_3$-Mg(1 wt%) | 67.1 | 89.8 | 0.48 |
| Ru/$Al_2O_3$-Na(1 wt%) | 67.6 | 90.9 | 0.50 |
| Ru/$Al_2O_3$-K(1 wt%) | 69.7 | 91.7 | 0.49 |

Even though the dry reforming reaction stoichiometry indicates an $H_2$/CO = 1, it is the water–gas shift inverse reaction (WGSR) responsible for the $H_2$/CO decrease up to 0.45 and 0.50 for the Ru catalyst series because WGSR ($CO_2$ + $H_2$ $\leftrightarrow$ CO + $H_2O$) consumes some of the hydrogen produced by dry reforming that reacts with $CO_2$ giving more CO and water.

### 2.2.3. Study of Ru Catalyst Activities Far from Equilibrium Conditions

In order to find differences between the catalytic performances of the different catalysts, it was necessary to work in conditions far from the equilibrium state, and for this reason, the catalyst weight in the experiments was reduced (50 mg). The results of these experiments are shown in Table 3. All the Ru/$Al_2O_3$-A/AE(1 wt%) catalysts display higher conversions than Ru/$Al_2O_3$ sample. With respect to the influence of the alumina dopant (A or AE), it can be seen that Ru catalysts supported on alumina doped with alkali metals (A), such as Na and K, gave the best catalytic results. In these catalysts, the presence of both alkali metals (Na or K) completely poisons all the acid sites of the alumina, as isopropanol dehydration experiments from a previous study showed [38]. On the other hand, the addition of AE metals partially inhibits the acidity of the support. Hence, it can be concluded that the acidity of the support modifies the metal-support interaction, thus improving the catalytic behavior. There would be no influence of the metallic dispersion since the average particle sizes of both the monometallic catalyst and the series doped with 1 wt% A/AE metal were similar, as Table 1 shows.

In addition, all Ru catalysts show a low $H_2$/CO molar ratio (between 0.38 and 0.46), indicating that Ru promotes WGSR (see Table 3).

**Table 3.** % $CH_4$ conversion, % $CO_2$ conversion, and $H_2$/CO molar ratio for Ru/$Al_2O_3$-A/AE(1 wt%) catalysts (750 °C, 135 min reaction time). Catalyst weight: 50 mg.

| Catalyst | X $CH_4$ (%) | X $CO_2$ (%) | Molar Ratio $H_2$/CO |
|---|---|---|---|
| Ru/$Al_2O_3$ | 34.1 | 59.2 | 0.41 |
| Ru/$Al_2O_3$-Ba(1 wt%) | 35.2 | 61.5 | 0.40 |
| Ru/$Al_2O_3$-Ca(1 wt%) | 42.6 | 67.7 | 0.46 |
| Ru/$Al_2O_3$-Mg(1 wt%) | 38.7 | 64.0 | 0.38 |
| Ru/$Al_2O_3$-Na(1 wt%) | 48.3 | 75.9 | 0.41 |
| Ru/$Al_2O_3$-K(1 wt%) | 56.9 | 83.2 | 0.46 |

### 2.2.4. Study of the Influence of Lower Ru Loading

If the noble metal loading could be decreased, the final cost of the catalyst would be lower. In this sense, for the sake of comparison, a Ru(0.5 wt%)/$Al_2O_3$-K(1 wt%) catalyst was compared with the Ru(1 wt%)/$Al_2O_3$-K(1 wt%) one. The results show that the higher the Ru loading, the higher $CH_4$ and $CO_2$ conversions, meaning that the lower Ru loading

(0.5 wt%) is not enough to attain good catalytic activities in the dry reforming of methane. A similar behavior was found in previous work [50] for Pt catalysts supported by doped alumina when the noble metal loading decreased from 0.5 wt% to 0.1 wt%.

### 2.2.5. Study of the A/AE Metal Loading Increase from 1 wt% to 5 or 10 wt%

Ru 1 wt% catalysts were used for this study since they gave the best activities with respect to Ru 0.5 wt% ones. The selected A/AE metal concentrations were 1, 5, and 10 wt%.

Figure 7 displays $CH_4$ and $CO_2$ conversions and $H_2/CO$ molar ratio obtained for the different Ru catalysts supported on $Al_2O_3$-A/AE(1, 5, and 10 wt%). All the Ru catalysts supported on alumina doped with 5 and 10 wt% A/AE metal showed much higher $CH_4$ and $CO_2$ conversions than the ones supported on $Al_2O_3$-A/AE(1 wt%). Values between 89 and 94% for $CO_2$ conversion and between 75 and 79% for $CH_4$ conversion were attained with catalysts supported on 10 wt% A/AE metal-doped alumina. Among all the catalysts, Ru/$Al_2O_3$-K(10 wt%) showed the best catalytic performance, with $CO_2$ and $CH_4$ conversions of 94 and 79%, respectively. This catalyst showed the smallest mean particle size (observed by TEM) of the series of Ru/$Al_2O_3$-A/AE (10 wt%) samples. Moreover, the elimination of the acidity and the development of basic sites on alumina (after A/AE addition at high concentrations) would favor the metal-support interactions leading to better catalysts for the dry reforming process.

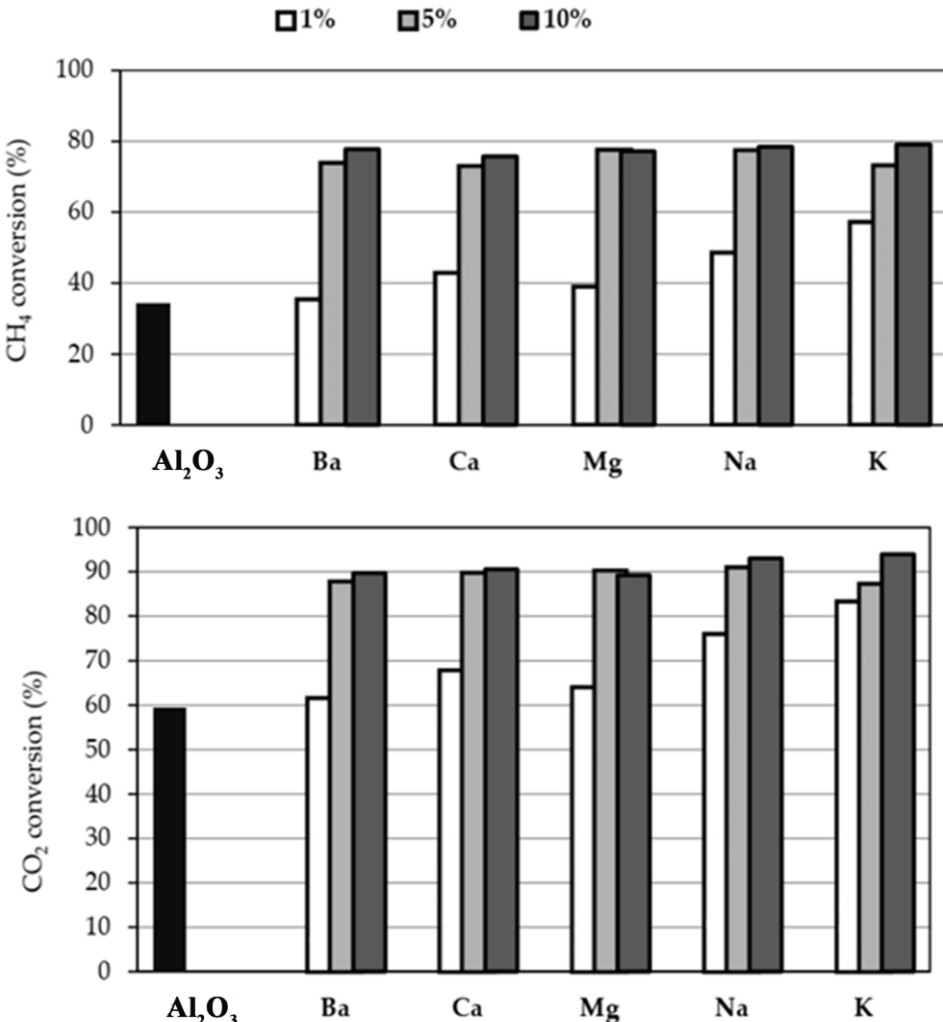

**Figure 7.** *Cont.*

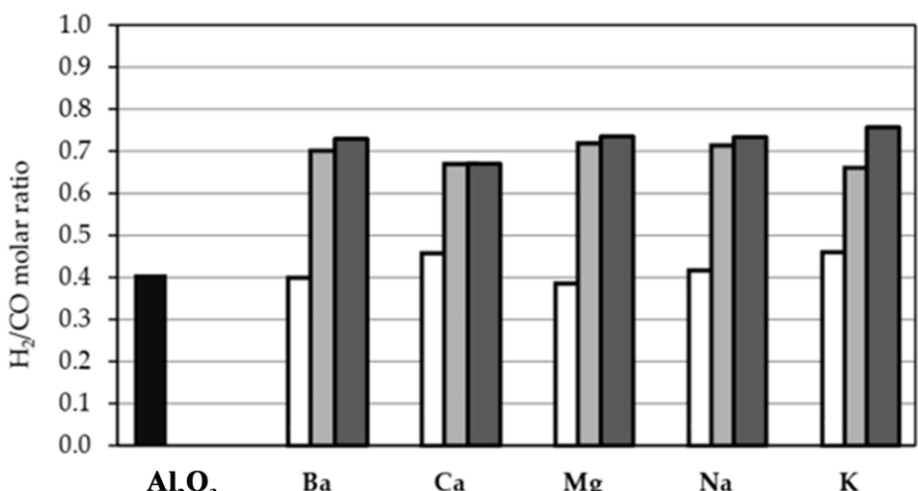

**Figure 7.** $CH_4$ conversion, $CO_2$ conversion, and $H_2/CO$ molar ratio for $Ru/Al_2O_3$-A/AE(1,5, and 10 wt%) catalysts (750 °C, 135 min reaction time).

$H_2/CO$ molar ratios were also highly increased (values between 0.66 and 0.73) when the A/AE metal loadings increased from 1 to 5 wt%, and they remained rather constant when the dopant increased from 5 to 10 wt% (see Figure 7). This indicates that for $Ru/Al_2O_3$-A/AE catalysts, the higher the dopant concentration, the less favored the WGSR, as was previously explained.

### 2.2.6. Catalyst Performance at Long Reaction Times Stability Test

The best catalyst, $Ru/Al_2O_3$-K(10 wt%), was chosen to be evaluated in a long dry reforming reaction during 59 h. Figure 8a shows the conversion of $CH_4$ and $CO_2$, while Figure 8b displays the $H_2/CO$ molar ratio obtained as a function of the long reaction time. This catalyst showed a short activation time, reaching a good and stable catalytic activity in 5 h. The average conversions of $CO_2$ and $CH_4$ were about 89 and 73%, respectively. With respect to the $H_2/CO$ molar ratios produced during the very long reaction (see Figure 8b), the values also remained practically constant (between 0.65 and 0.70). Hence, the stability of the catalyst was excellent during the reaction time. Taking into account that the deactivation of the metallic phase of these catalysts could be caused by carbonaceous deposits and/or sintering of the active phase [51], additional TPO and TEM experiments were carried out. According to the TPO test, the carbon content of the used catalyst was negligible (<0.1 wt%), being resistant to deactivation by carbon deposition during long reaction times at high temperatures. It seems that there would be a continuous cleaning-up mechanism. In doped alumina-supported ruthenium catalysts, there would be a continuous cleaning mechanism that removes the carbon formed during methane decomposition so that the metallic surface remains coke-free during a long time reaction (59 h) at 750 °C. This mechanism would sweep away the carbon formed by methane decomposition leaving the metallic surface free of carbon even at long reaction times. Previous studies [34] have demonstrated that on similar supported noble metal catalysts, the $CH_4$ decomposition and the $CO_2$ dissociation occur via two independent paths. The first path involves the decomposition of $CH_4$ on the metal particle, resulting in the formation of $H_2$. The carbon formed during the decomposition of $CH_4$ can deposit carbon on the metal. The second path involves the dissociation of $CO_2$ on the support (but near the metal particle) to form CO and O. The oxygen formed during the dissociation can then reoxidize the support to provide a redox mechanism for continuous cleaning of the carbon deposited on the metal. The balance between the rate of decomposition and the rate of cleaning determines the overall stability of the catalyst.

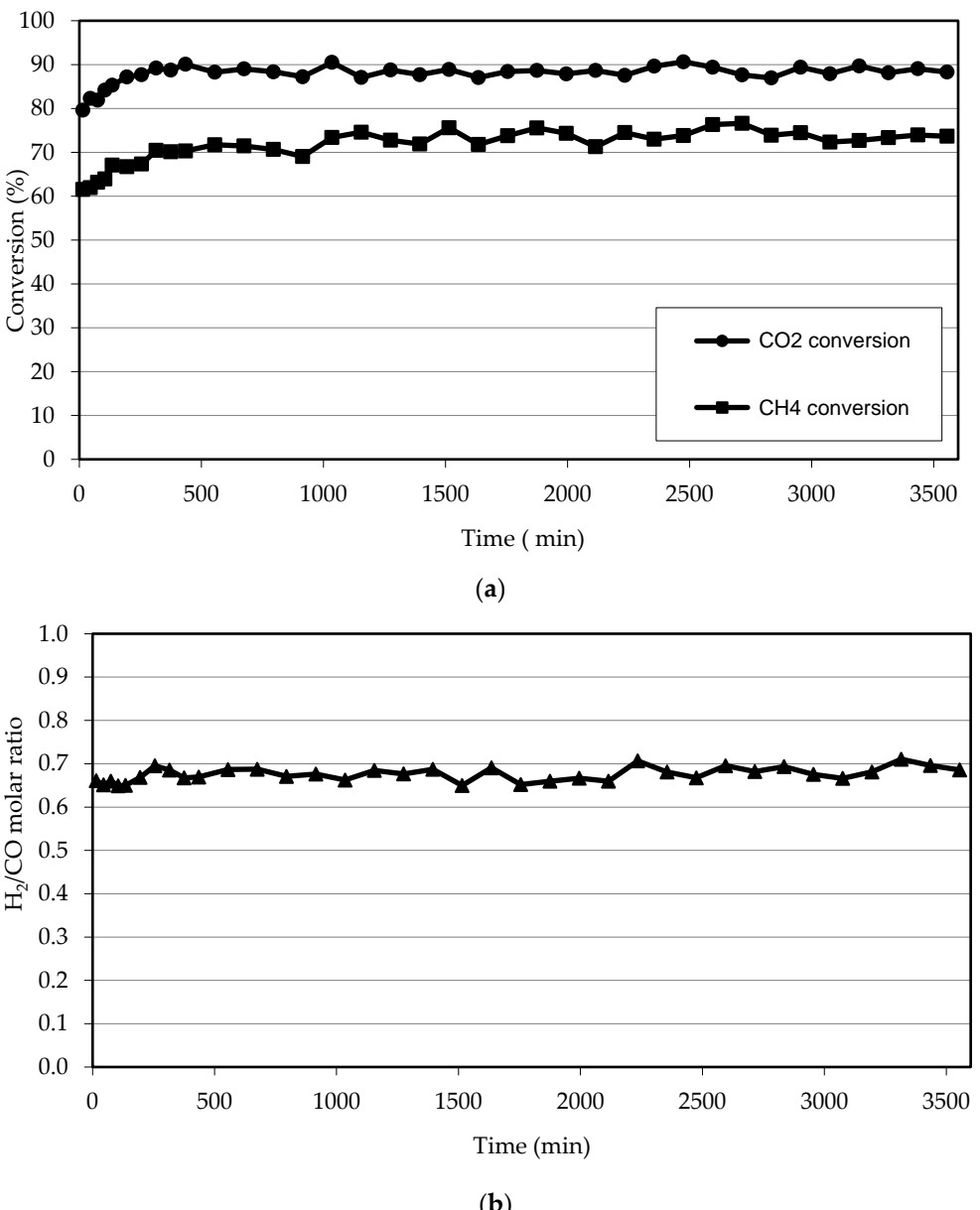

**Figure 8.** $CO_2$ and $CH_4$ conversions (**a**) and $H_2/CO$ molar ratios (**b**) vs. reaction time for Ru(1 wt%)/$Al_2O_3$-K(10 wt%) catalyst tested during 59 in the dry reforming reaction at 750 °C.

Considering TEM results, when comparing fresh and used (after 59 h reaction time at 750 °C) samples, it can be seen that the particle sizes of the fresh catalyst increased from 2.8 nm to 5.8 nm. This could indicate that even though Ru catalysts are resistant to coke deposition, the metallic particles were sintered during long reaction times at high temperatures. The sinterization process could lead to a catalyst deactivation at much longer reaction times, thus probably affecting the activity.

For the sake of comparison of the catalytic activity, a very well-studied [52–56] Pt(1 wt%)/$Al_2O_3$ catalyst was chosen to be tested in the same conditions of the dry reforming reaction at 750 °C and 59 h. During the first hours of the reaction, $CH_4$ and $CO_2$ conversions were very high (72 and 91%, respectively); then, the catalyst showed an important activity loss, the conversions decreasing up to 20 and 37%, respectively. TPO experiments of the Pt/$Al_2O_3$ catalyst showed that one of the reasons for this deactivation process could be the carbon deposition since the coke content of the used catalyst was 0.3 wt%. Other authors [56] found the same activity loss caused by coke deposition for this



reaction during long reaction times. On the other hand, by comparing with Ir catalysts [35], (Ir(1 wt%)/Al$_2$O$_3$-Mg(10 wt%)) also showed excellent stability during long reaction times (59 h) that gave 93.5% and 81.2% for CO$_2$ and CH$_4$ average conversions, respectively. Likewise, other authors in recent years evaluated Ru catalysts in long-term dry reforming reactions. Singh and Madras [40] synthesized Ru catalysts supported on TiO$_2$ (with very high metallic content, 4% wt%) and found them to be highly active for this reforming reaction at 650 °C, but the stability of the catalysts was not good throughout only 20 h of reaction. Liu et al. [41] showed good stability (but with a 10% conversion loss for 25 h reaction time) for the Ru(0.5 wt%))-CeO$_2$ catalyst at 700 °C. Finally, Li et al. [42] found superior stability of Ru/Mg$_3$(Al)O catalyst for a 300 h of a long-term catalytic test, though the metallic charge of Ru was very high (2 wt%).

From these results, it is evident that good activity, stability, and H$_2$/CO molar ratios of Ru catalysts supported doped Al$_2$O$_3$ for the dry reforming process.

## 3. Materials and Methods

### 3.1. Preparation of the Catalysts and Supports

Catalysts were prepared using commercial γ-Al$_2$O$_3$ (Cyanamid Ketjen CK-300, Wayne, NJ, USA), SBET = 190 m$^2$ g$^{-1}$ and Vpore = 0.5 cm$^3$ g$^{-1}$. Alkaline (Na or K) and alkaline earth metals (Ba, Ca, or Mg) were deposited on the support by using 1M aqueous solutions of NaOH, KOH, Ba(NO$_3$)$_2$, Ca(NO$_3$)$_2$, or Mg(NO$_3$)$_2$ with a solution volume/support ratio of 1.4 mL g$^{-1}$ to give a final loading of 1, 5, and 10 wt%. The modified supports were calcined in 50 mL min$^{-1}$ air flow at 800 °C for 3 h. Incipient wetness impregnation was achieved at room temperature using RuCl$_3$.3H$_2$O solution with concentrations suitable for attaining final Ru loading of 1 wt%.

Finally, the catalysts were calcined in 50 mL min$^{-1}$ air flow at 500 °C for 3 h. The catalysts were named as follows: Ru/A$_{12}$O$_3$-A/AE(loading as wt%) being A: alkaline metal and AE: alkaline earth metal.

### 3.2. Characterization of the Catalysts and Supports

XRD patterns, textural properties, temperature-programmed desorption (TPD) of CO$_2$ analyses, and the 2-propanol dehydration reaction were previously studied by Maina et al. [38] for the different supports of alumina doped with alkaline and alkaline earth metals.

TPR experiments were performed in homemade equipment by heating the sample at 6 °C min$^{-1}$ under a reductive mixture of H$_2$ (5% *v/v*)/N$_2$ (9 mL min$^{-1}$). The catalysts were previously oxidized in situ in airflow at 500 °C for 3 h.

XPS determinations were made with a Photoemission Electron Spectrometer equipped with an X-ray source Mg/Al and a hemispherical analyzer PHOIBOS 150 in the fixed analyzer transmission mode (FAT). The spectrometer operates with an energy power of 100 eV, and the spectra were obtained with a pass energy of 30 eV and an Mg/Al anode operated at 200 W. Samples were previously reduced under H$_2$ at 750 °C for 5 h in a flow reactor and then placed in the equipment and reduced in situ with H$_2$ at 400 °C for 10 min. The binding energies (BE) were referred to as the C1s peak at 284 eV. Peak areas and BE values were estimated by fitting the curves with the combination of Lorentzian–Gaussian curves of variable proportion using the Casa XPS Peak-fit software 1.2.

TEM measurements were carried out in a JEOL 100 CX microscope (Tokyo, Japan) with a nominal resolution of 6 Å, operating with an acceleration voltage of 100 kV and magnification ranges of 80,000× and 100,000×. The samples, previously reduced under H$_2$ at 750 °C for 5 h in a flow reactor, were prepared by grinding, suspension, and sonication in ethanol, placing a drop of the suspension on a carbon copper grid. After evaporating the solvent, the specimens were placed into the microscope column. With respect to the size distributions of the metallic Ru particles and the measurements of the average sizes determined by TEM, they were carried out by an expert in Transmission Electron Microscopy techniques. For each catalyst, an average of 100 metallic particles was analyzed.

The mean particle diameter (d) was calculated as $d = \Sigma\, n_i d_i\, /\Sigma\, n_i$, where $n_i$ is the frequency of particles with diameter $d_i$.

TPO experiments were performed in a homemade instrument to determine the amount of carbon deposited on the catalyst surface after methane dry reforming. The test was carried out in a continuous flow reactor at atmospheric pressure using 40–60 mg of spent catalyst. This sample was first oxidized with a mixture of $O_2/N_2$ 5% *v/v* at 60 cm$^3$ min$^{-1}$. The oxidation was achieved at 10 °C min$^{-1}$ from room temperature to 650 °C. The outlet gases were fed into the reactor to be converted into methane over a Ni/Kieselgur catalyst in a hydrogen atmosphere. The gases at the outlet of the reactor were analyzed by FID gas chromatograph.

*3.3. Catalytic Tests: Methane Dry Reforming Reaction*

The dry reforming of methane (DRM) was carried out in flow equipment at 750 °C for 135 min. The samples (50 mg) were first reduced under flowing $H_2$ (30 mL min$^{-1}$) at 750 °C for 5 h. After reduction, He was passed through the catalyst bed for 15 min, and finally, the $CH_4/CO_2$ mixture ($CH_4/CO_2$ molar ratio = 1) was fed into the reactor with a flow rate of 20 mL min$^{-1}$. Diffusion effects were found to be absent due to the small catalyst particles. The reaction products were analyzed using a Varian Star 3400 CX gas chromatograph with a Supelco Carboxen 1006 PLOT (30 m × 0.53 mm) column and a TCD detector. In order to study the catalyst stability, additional experiments at 750 °C were performed for longer reaction times (59 h). The conversions of $CH_4$ and $CO_2$ and $H_2/CO$ ratio were calculated according to the Formulas (1)–(3) below:

$$CH_4 \text{ conversion}: \ x_{CH_4} = \frac{\left(F_{CH_4\ in} - F_{CH_4\ out}\right)}{F_{CH_4\ in}} \times 100 \tag{4}$$

$$CO_2 \text{ conversion}: \ x_{CO_2} = \frac{\left(F_{CO_2\ in} - F_{CO_2\ out}\right)}{F_{CO_2\ in}} \times 100 \tag{5}$$

where $FCO_2$ in and $FCH_4$ in are the inlet flows for $CH_4$ and $CO_2$, and $FCO_2$ out and $FCH_4$ out in are the outlet flows for $CH_4$ and $CO_2$, respectively.

$$H_2/CO \text{ ratio}: \ H_2/CO = \frac{n_{H_2}}{n_{CO}} \tag{6}$$

Being:

$$n_{H_2} = \frac{1.76\, A_{H_2}}{M_{H2}} = \frac{1.76\, A_{H_2}}{2} \tag{7}$$

$$n_{CO} = \frac{0.67\, A_{CO}}{M_{CO}} = \frac{067\, A_{CO}}{28} \tag{8}$$

$n_j$: moles of species j
$A_j$: area of species j (obtained from the chromatogram)
$M_j$: molecular weight of species j

## 4. Conclusions

Very good catalytic behavior in the dry reforming reaction was shown by catalysts supported on 1 wt% Na or K-doped alumina that displayed a negligible acidity that modifies the metal-support interaction and the catalytic performance.

The higher the A/AE metal concentration (from 1 to 5 and 10 wt%), the higher the $CO_2$ and $CH_4$ conversions for all Ru catalysts and the $H_2/CO$ ratios. Among all the catalysts, $Ru/Al_2O_3$-K(10 wt%) catalyst showed the best catalytic performance, in agreement with the smallest particle size that it has as compared with the series of $Ru/Al_2O_3$-A/AE(10 wt%) samples. Moreover, the elimination of the acidity and the development of basic sites on alumina would favor the metal-support interactions, thus leading to better catalysts.

TPR and XPS results showed that the main percentage of Ru was in a zerovalent state, the rest being in an oxidized state. Ru in catalysts supported on 10 wt%-doped alumina would have a higher concentration of $Ru^0$, and this would be one of the reasons for the increase of the catalytic activity with respect to 1 wt%-doped alumina ones.

As TEM results indicated, 1 wt%-doped alumina Ru catalysts showed smaller and narrower distributed metallic particles than 10 wt%-doped alumina ones. In the first group, the Ru reducibility would be incomplete, so the catalytic activity was lower. In the second group, Ru was highly reduced, and the particle sizes were larger, so an important influence of the metal-support interaction in the catalytic performance could be expected.

It is worth mentioning that the selected catalyst (Ru/Al$_2$O$_3$-K(10 wt%)) displayed a very good catalytic activity, without any activity fall during an experiment of 59 h-reaction time. Ru metallic phase showed to be resistant to coke formation even though its particles were sintered at the longer reaction.

**Author Contributions:** Conceptualization, S.C.P.M., A.D.B., I.M.J.V. and S.R.d.M.; formal analysis, S.C.P.M., A.D.B., I.M.J.V. and S.R.d.M.; funding acquisition, A.D.B. and S.R.d.M.; investigation, S.C.P.M., A.D.B. and S.R.d.M.; methodology, S.C.P.M., A.D.B., I.M.J.V. and S.R.d.M.; project administration, A.D.B. and S.R.d.M.; supervision, S.C.P.M., A.D.B. and S.R.d.M.; validation, S.C.P.M., A.D.B. and I.M.J.V.; writing—original draft, S.C.P.M., I.M.J.V. and S.R.d.M.; writing—review and editing, S.C.P.M., A.D.B., I.M.J.V. and S.R.d.M. All authors have read and agreed to the published version of the manuscript.

**Funding:** This research was funded by CONICET (PIP 11220170100002CO) and Universidad Nacional del Litoral (Project CAI+D PI 50320220100088LI)—Argentina.

**Acknowledgments:** Authors thank G.M. Baez for experimental assistance. This work was carried out with the financial support of Universidad Nacional del Litoral (Project CAI+D) and CONICET (Project PIP), Argentina.

**Conflicts of Interest:** The authors declare no conflict of interest.

## Appendix A

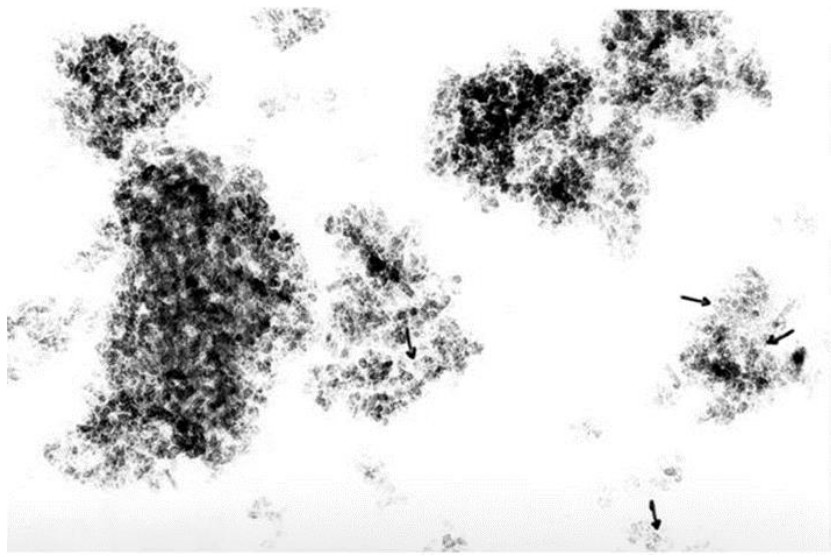

**Figure A1.** Microphotography of Ru/Al$_2$O$_3$ catalyst.

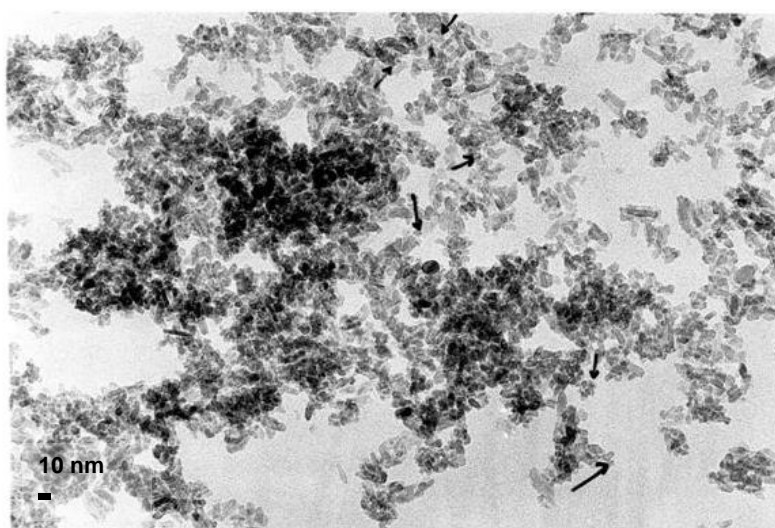

**Figure A2.** Microphotography of Ru/Al$_2$O$_3$-Na(1 wt%) catalyst.

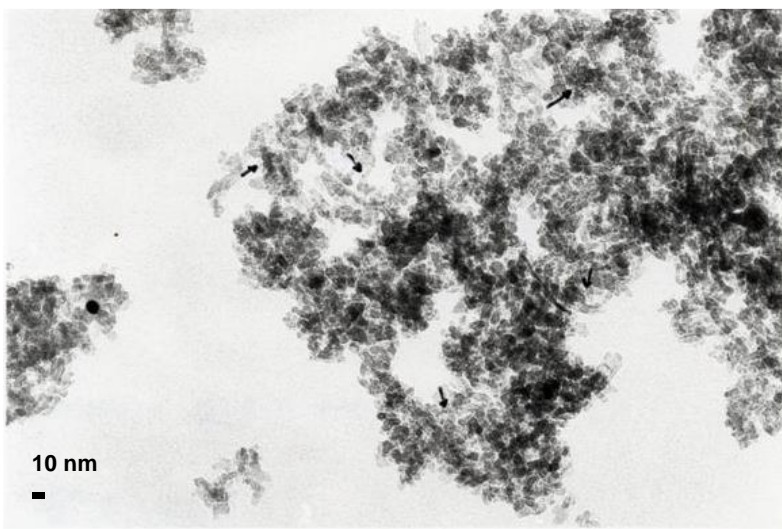

**Figure A3.** Microphotography of Ru/Al$_2$O$_3$-K(1 wt%) catalyst.

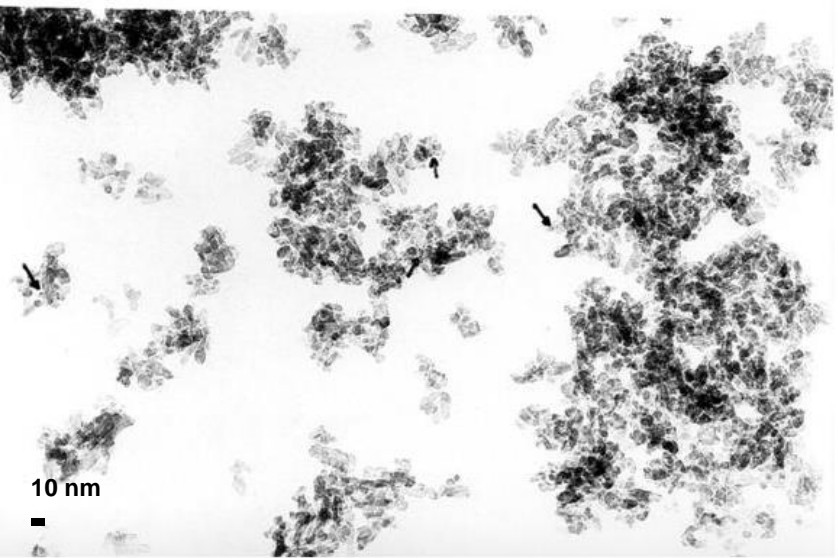

**Figure A4.** Microphotography of Ru/Al$_2$O$_3$-Mg(1 wt%) catalyst.

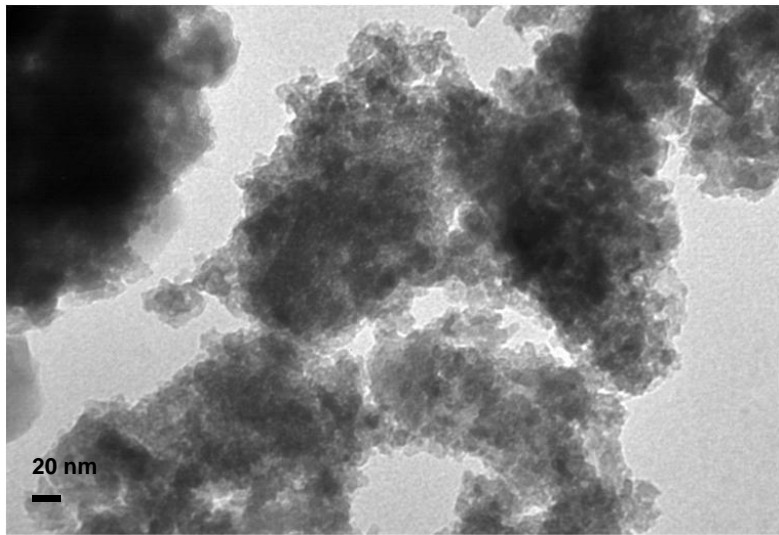

**Figure A5.** Microphotography of Ru/Al$_2$O$_3$-Na(10 wt%) catalyst.

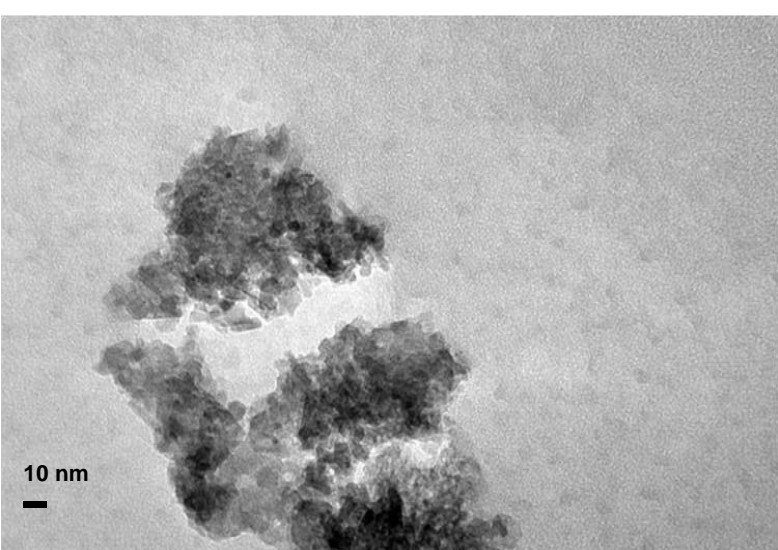

**Figure A6.** Microphotography of Ru/Al$_2$O$_3$-K(10 wt%) catalyst.

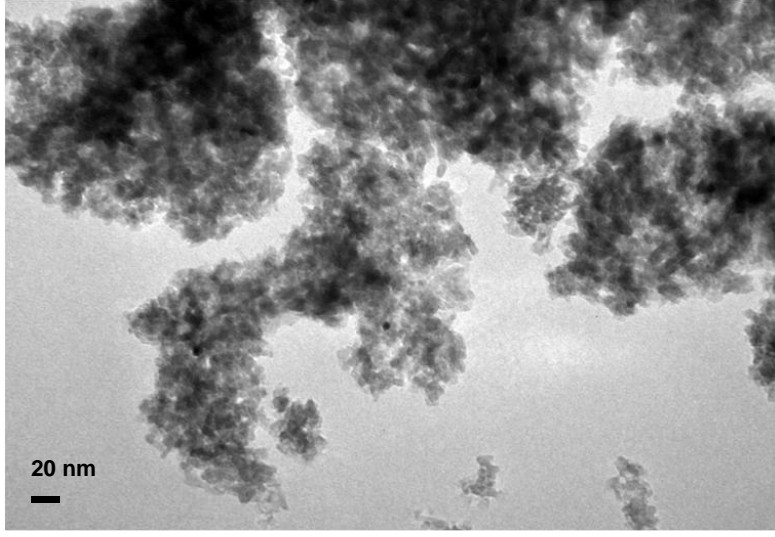

**Figure A7.** Microphotography of Ru/Al$_2$O$_3$-Ba(10 wt%) catalyst.

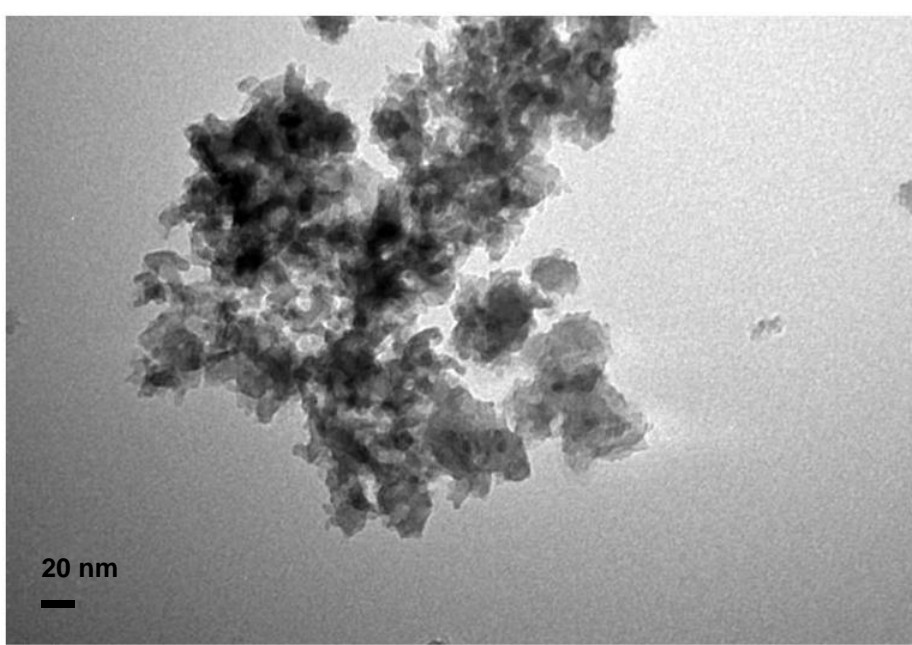

**Figure A8.** Microphotography of Ru/Al$_2$O$_3$-Ca(10 wt%) catalyst.

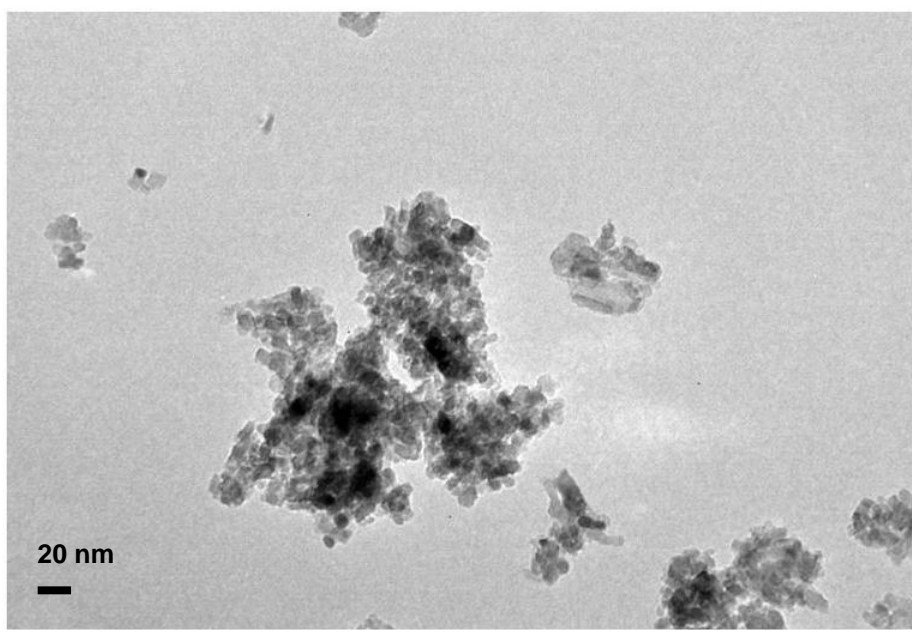

**Figure A9.** Microphotography of Ru/Al$_2$O$_3$-Mg(10 wt%) catalyst.

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
