# Peer review of "Performance of Modified Alumina-Supported Ruthenium Catalysts in the Reforming of Methane with CO2"

_catalysts, doi:10.3390/catal13020338_

Round 1

Reviewer 1 Report

In this work, the alumina doped with alkaline (Na and K) and alkaline earth metals (Ba, Ca and Mg) were prepared and used to support Ru (1 wt%) catalysts. The supports and catalysts were characterized by various techniques, and the catalysts were studied for the dry reforming of methane. The manuscript was well prepared, however, there are still are questions need to be addressed. And I recommend a major revision before acceptance.

(1).   It was said that The Ru catalyst seemed to be resistant to coke formation even though its particles are sintered during a long-term reaction. I suggest to test the content of the carbon deposition from the spent catalysts.

(2).   For the TPR profiles in Fig. 1 and Fig. 2, the loading of Ru was only 1%, while the reduction peaks were relatively large. How about the TPR profile for alumina support without Ru.

(3).   For the Ru3p3/2 XPS signals in Fig. 4, I believe there are some RuOx in the sample, which should be deconvoluted.

(4).   Some of the Fig.s can be merged into one Fig., such as the Figs. 5 and Figs. 6.

(5).   For the Fig. 5 and Fig. 6, how can the particle size be identified from the TEM images. Moreover, it should be pointed out what these particles are? Are they Ru particles or Al2O3 particles?

(6).   For the catalytic performance, how about the selectivity of H2 and CO, not just list the molar ratio of H2/CO.

(7).   The conclusions are too long, it should be further condensed.  

Author Response

Dear Reviewer,

I am sending you the response and the revised paper titled: “Performance of modified alumina-supported ruthenium catalysts in the reforming of methane with CO2” by Silvia C.P. Maina, Julieta I. Vilella Adriana D. Ballarini and Sergio R. de Miguel, which was corrected taking into account the reviewers suggestions. Besides, the answers to the reviewers are included.

Sincerely yours.

Reviewer 2 Report

In this paper, the catalytic performance of (1, 5 and 10 wt%) Ba, Ca, Mg, Na and K modified Al2O3-supported Ru catalyst for CH4/CO2 reforming reaction was studied. With K modification, the Ru/Al2O3-K (10 wt%) showed long-time catalytic stability for 59 hours (750℃), in which a high conversion rate of CH4/CO2 was achieved, and the formation of carbon deposit was well inhibited. The following suggestions were provided for further improving the quality of the manuscript. If this manuscript could be revised carefully, it might be considered as a suitable article for publication in the journal of catalysts. These suggestions were proposed as follows.

1. The method used for catalyst preparation is not mentioned in the abstract, and the specific reaction time should be written in "long-time dry reforming reactions" in line 16.

2. This paper demonstrates the excellent catalytic performance of this catalyst for CH4/CO2 reforming reaction. However, the calculation process, including CO2 and CH4 conversion as well as H2/CO selectivity, should be introduced.

3. For the catalytic reforming of methane, apart from these catalytic reactions of the modified Al2O3-supported Ru, the related catalysts reported in recent three years should be also introduced to compare with these catalysts studied in this paper so as to highlight the innovation of this research.

4. By comparison, the Ru/Al2O3-K(10 wt%) catalyst exhibited the best catalytic activity. In light of the good property, the structure-activity relationship should be revealed in detail.

5. The modified Al2O3-supported Ru catalyst has a good carbon deposition removal effect after 59 hours of reaction, thus prolonging the service life of the catalyst. Please briefly describe the mechanism of carbon deposition removal.

Author Response

(The authors gave the same response as above.)

Round 2

Reviewer 1 Report

The authors have replied the comments, I think it can be accepted.

Reviewer 2 Report

 These key issues raised has been revised clarified. Thus, the manuscript is suggested to be published on this journal of Catalysts.